# Effects of Insertion of Ag Mid-Layers on Laser Direct Ablation of Transparent Conductive ITO/Ag/ITO Multilayers: Role of Effective Absorption and Focusing of Photothermal Energy

**DOI:** 10.3390/ma14185136

**Published:** 2021-09-07

**Authors:** Younggon Choi, Hong-Seok Kim, Haunmin Lee, Wonjoon Choi, Sang Jik Kwon, Jae-Hee Han, Eou-Sik Cho

**Affiliations:** 1Department of Electronic Engineering, Gachon University, 1342 Seongnam-daero, Sujeong-gu, Seongnam-si 13120, Gyeonggi-do, Korea; cyg1994@gmail.com (Y.C.); sjkwon@gachon.ac.kr (S.J.K.); 2Department of Materials Science and Engineering, Gachon University, 1342 Seongnam-daero, Sujeong-gu, Seongnam-si 13120, Gyeonggi-do, Korea; hskim2024@gmail.com; 3School of Mechanical Engineering, Korea University, 145 Anam-ro, Seongbuk-gu, Seoul 02841, Korea; skekgusals@korea.ac.kr (H.L.); wojchoi@korea.ac.kr (W.C.)

**Keywords:** ITO/Ag/ITO multilayer, laser direct ablation, Nd:YVO_4_ laser, thermal conductivity, photothermal energy, transparent conductive oxide electrode

## Abstract

From the viewpoint of the device performance, the fabrication and patterning of oxide–metal–oxide (OMO) multilayers (MLs) as transparent conductive oxide electrodes with a high figure of merit have been extensively investigated for diverse optoelectronic and energy device applications, although the issues of their general concerns about possible shortcomings, such as a more complicated fabrication process with increasing cost, still remain. However, the underlying mechanism by which a thin metal mid-layer affects the overall performance of prepatterned OMO ML electrodes has not been fully elucidated. In this study, indium tin oxide (ITO)/silver (Ag)/ITO MLs are fabricated using an in-line sputtering method for different Ag thicknesses on glass substrates. Subsequently, a Q-switched diode-pumped neodymium-doped yttrium vanadate (Nd:YVO_4_, λ = 1064 nm) laser is employed for the direct ablation of the ITO/Ag/ITO ML films to pattern ITO/Ag/ITO ML electrodes. Analysis of the laser-patterned results indicate that the ITO/Ag/ITO ML films exhibit wider ablation widths and lower ablation thresholds than ITO single layer (SL) films. However, the dependence of Ag thickness on the laser patterning results of the ITO/Ag/ITO MLs is not observed, despite the difference in their absorption coefficients. The results show that the laser direct patterning of ITO/Ag/ITO MLs is primarily affected by rapid thermal heating, melting, and vaporization of the inserted Ag mid-layer, which has considerably higher thermal conductivity and absorption coefficients than the ITO layers. Simulation reveals the importance of the Ag mid-layer in the effective absorption and focusing of photothermal energy, thereby supporting the experimental observations. The laser-patterned ITO/Ag/ITO ML electrodes indicate a comparable optical transmittance, a higher electrical current density, and a lower resistance compared with the ITO SL electrode.

## 1. Introduction

Transparent conductive oxides (TCOs) such as indium tin oxide (ITO), indium zinc oxide, and aluminum-doped zinc oxide films have been used extensively as pixel electrodes and window layers in various optoelectronic and energy devices, including flat panel displays (FPDs), touch screen panels (TSPs), and solar cells [1,2,3,4,5,6,7,8,9,10,11,12,13]. Recently, conventional metal electrodes such as Mo, Cu, Al, and Ag paste films have been replaced with TCO materials of higher optical transmittance to realize fully transparent optoelectronic devices. However, replacing the metal electrodes with pure TCOs in large-sized FPDs or TSPs is not optimal because of their lower conductivities compared with those of metals. Therefore, various oxide–metal–oxide (OMO) multilayers (MLs) have been considered and investigated as candidate replacements exhibiting of both high optical transparency and electrical conductivity [14,15,16,17,18,19,20,21,22]. Despite the insertion of thin metal mid-layers between TCO films, the transparency of OMO MLs did not reduce significantly, whereas the electrical conductivities improved compared with those of pure TCO [23,24,25]. In addition, the thickness of the metal insertion layer can be increased to improve the electrical conductivity because its optical transmittance does not decrease monotonically with thickness but oscillates periodically as a function of thickness [26]. In particular, it has been reported that an optimized ITO/Ag/ITO ML possesses a high figure of merit owing to its high optical transmittance and low electrical resistivity [25,27]. The Ag has a much lower resistivity of 1.59 μΩ cm than those of other materials such as Cu, Al, and Mo. In terms of optoelectronics, the energy band of d-orbital in Ag is located below Fermi energy level (E_F_) and less photons are expected to be absorbed into Ag in the visible spectrum. Therefore, a higher transmittance of Ag material is expected in a wavelength of visible ray regime [28]. Recently, Wang et al. successfully reported the inorganic all-solid-state electrochromic device applications based on ITO/Ag/ITO ML electrodes with a high figure of merit [25]. The role of Ag insertion into other semiconducting matrices has been reported for control of thermal conductivity of the system [29,30]. However, to the authors’ best knowledge, there has been no study on the underlying mechanism by which Ag mid-layers play an important role of effective absorption and focusing of photothermal energy for patterning the ITO/Ag/ITO ML electrodes by using laser direct ablation.

To maintain the increasing demand for fully transparent electrodes, repetitive, time- and cost-effective, and reliable patterning techniques must be developed for forming OMO ML electrodes. It is difficult to use photolithography for the patterning of OMO ML electrodes because of the different etching properties of the oxide and metal layers. The laser direct ablation technique has been applied to the patterning of ITO layered electrodes [31,32,33] and ITO/Ag/ITO MLs [34] because of its reliability and simplicity. In addition, the utilization of this technique has also been successfully demonstrated for patterning of other metallic nanomaterials and oxide layers, including Ag nanowires, ZnO, Ni/NiO hybrid electrodes, and NiOx thin films. [35,36,37,38]. As the inserted Ag mid-layer absorbs the laser beam, the ITO/Ag/ITO ML is effectively melted and removed by direct laser patterning. In this study, to analyze the effect of the inserted Ag mid-layer on the etching of ITO/Ag/ITO MLs, the latter were deposited on glass substrates with Ag insertion layers of different thicknesses using an in-line sputtering method. A Q-switched diode-pumped neodymium-doped yttrium vanadate (Nd:YVO_4_; 1064 nm wavelength) laser was used for the direct ablation of the prepared ITO/Ag/ITO MLs. The experimental samples with different Ag thicknesses were compared as a function of the various laser beam conditions. The effects of the inserted Ag mid-layer on the laser direct patterning of the ITO/Ag/ITO MLs were investigated and analyzed. The experimental results presented herein were supported by simulation results; hence, we expect to obtain optimal processing conditions for the laser direct patterning of ITO/Ag/ITO ML thin films such that they are applicable to high-performance optoelectronic and energy devices.

## 2. Materials and Methods

### 2.1. Fabrication of ITO/Ag/ITO ML and ITO Single Layer (SL) Substrates

A soda-lime glass of 1 mm thickness was used as a substrate. After cleaning the glass substrate ultrasonically in the ultrasonic bath (Branson 5510EDTH) that was operated at a frequency of 60 Hz with a voltage of 220V, a bottom ITO layer was deposited on the prepared substrates via direct current (DC) magnetron sputtering at a power density of 2.24 W/cm^2^. The thickness of the deposited ITO layer was measured to be 53.68 ± 0.16 nm. Subsequently, an Ag mid-layer was deposited on the bottom ITO layer via radio frequency (RF) magnetron sputtering at a power density of 1.16 W/cm^2^ by controlling the sputtering time to 0 (Ag mid-layer not inserted), 30, 60, and 90 s to obtain Ag layers of different thicknesses. For the measurement of Ag thickness, Ag layers were deposited on bare glasses during RF sputtering on the bottom ITO layer under the same processing conditions. An ML structure was finally achieved by depositing the top ITO layer on the already-formed glass/ITO/Ag substrate under the same processing conditions by which the bottom ITO layer was deposited. During the DC sputtering process for the formation of ITO layers, the operating pressure was maintained at 4 mTorr by injecting Ar and O_2_ gases into an in-line sputtering reactor at mass flows of 50 and 1.2 sccm, respectively. The speed of jig fixing a substrate was set at 105 cm/min (35 Hz), and a two-scan repetition was performed. For Ag deposition, an Ar flow of 15 sccm was injected into the reactor at an operating pressure of 3 mTorr. For Ag deposition times of 30, 60, and 90 s, the Ag thickness was measured to be 6, 13, and 16 nm, respectively.

### 2.2. Laser Direct Ablation Process

A Q-switched neodymium-doped yttrium vanadate (Nd:YVO_4_) pumped laser (Miyachi, ML-7111A, Miyachi Korea Co., Sungnam, Gyeonggi-do, Korea) was used to ablate the prepared ITO/Ag/ITO MLs. The solid-state laser system operated at a single TEM00 mode has a beam spot diameter of about 80 μm with a F-theta-160-mm-focal-length lens. The pulse duration and output optical wavelength of the laser were set to 10 ns and 1064 nm, respectively. The duty cycle of the lasing ranged from 12.5 to 50 μs, and the pulse energy ranged from 51 to 163 μJ. Figure 1 shows a schematic illustration of the laser system used in this study. For different laser pulse energies, the ITO/Ag/ITO MLs were etched using a laser system at various scanning speeds from 50 to 2000 mm/s.

### 2.3. Optical, Electrical, and Elemental Characterizations

The transmittance and sheet resistivity were measured using an ultraviolet–visible light spectrometer (Cary 100 UV-Vis, Agilent Technologies, Inc., Santa Clara, CA, USA) and a four-point probe (CMT-SR2000N, AIT Co., Ltd., Suwon, Gyeonggi-do, Korea) to characterize the ITO/Ag/ITO MLs. After performing direct laser ablation, the shapes and critical dimensions of the etched patterns were investigated using a high-resolution scanning electron microscope (Hitachi S-4700, Hitachi Korea Ltd., Seoul, Korea) and a surface profiler (Alpha-Step 500, KLA Tencor, Hwaseong, Gyeonggi-do, Korea). In addition, energy dispersive X-ray spectroscopy (EDX) was performed to investigate the presence of residue in the etched patterns. The electrical characteristics of the laser-ablated electrodes were obtained using a parameter analyzer (4156C Precision Semiconductor Parameter Analyzer, Agilent Technologies, Inc., Santa Clara, CA, USA).

### 2.4. Computational Simulation

Temperature distribution was analyzed using the heat transfer module of comsol multiphysics [39,40]. The details of the simulation conditions, such as the material properties, laser beam design (Gaussian beam), and specific set parameters of the materials used in the simulation, are provided in the Appendix A.

## 3. Results and Discussion

### 3.1. Experimental Aspects

Figure 2 shows the electrical and optical characteristics of the prepared ITO/Ag/ITO MLs, where the sheet resistances, in units of ohm per square (Ω/sq), of both the ITO/Ag/ITO MLs and Ag SLs as a function of Ag thickness are shown in Figure 2a. The MLs exhibited lower sheet resistances than the Ag SLs. As the Ag thickness increased from 0 to 16 nm, the sheet resistance decreased monotonically; the lower sheet resistance of the ITO/Ag/ITO MLs was attributed to the higher conductivity of Ag compared with those of conventional ITO bulk electrodes. The electrical resistances of the ITO/Ag/ITO ML, *R*_ITO/Ag/ITO_ were measured to be 49.06, 16.21, 8.03, and 3.52 Ω/sq, respectively, for Ag mid-layer thicknesses of 0, 6, 13, and 16 nm, respectively. The sheet resistance of *R*_ITO/Ag/ITO_ was lower than that of the Ag SL with the same thickness because the *R*_ITO/Ag/ITO_ of the ML configuration was due to the parallel combination of the sheet resistances of both the upper and lower ITO layers and the inserted Ag mid-layer.

Figure 2b shows the optical transmittance of the ITO/Ag/ITO MLs as a function of wavelength. In the visible light range from 400 to 800 nm, the average transmittances of the ITO/Ag/ITO MLs were 78.80%, 73.24%, 65.79%, and 43.89% for the thicknesses of the inserted Ag mid-layers of 0, 6, 13, and 16 nm, respectively. In the case of Ag SLs, average transmittances of 56%, 37%, and 19% were obtained for Ag thicknesses of 6, 13, and 16 nm, respectively. At a wavelength of 1064 nm, the transmittances of the ITO/Ag/ITO MLs were 76.59%, 61.87%, 38.07%, and 16.53% for the Ag mid-layers thicknesses of 0, 6, 13, and 16 nm, respectively. From the results of electrical and optical characteristics of Figure 2, the figure of merit (*FoM*) for each ITO/Ag/ITO ML was calculated using the equation defined as
(1)$$FoM=\frac{Trans^{10}}{R_{s}}$$
where *Trans* is the average transmittance in the wavelength from 400 to 800 nm in Figure 2a, and *R_S_* the sheet resistance of Figure 2b. The *FoM* of the ITO/Ag/ITO MLs were calculated as 1.88 × 10^−3^, 2.74 × 10^−3^, 1.89 × 10^−3^, and 7.54 × 10^−5^ Ω^−1^ for the thickness of the inserted Ag mid-layers of 0, 6, 13, and 16 nm, respectively. From the results, the insertion of a 6 nm Ag mid-layer indicates the optimal one regarding the *FoM* for each Ag mid-layer in this work.

For both Ag and ITO/Ag/ITO, it was observed that the optical transmittance decreased as the Ag thickness increased because the absorption of light energy increased [41]. Therefore, more laser beams are expected to be absorbed in the ITO/Ag/ITO ML structure as the thickness of Ag increases. The higher optical transmittance of the ITO/Ag/ITO ML may be due to the surface plasmon resonance arising from the interaction between ITO and Ag layers [42,43]. The optical transmittances of the ITO SL and ITO/Ag (6 nm)/ITO ML in the visible light regime did not differ significantly.

Figure 3 shows the laser direct ablation threshold of the ITO/Ag/ITO MLs. The relationship between the ablated spot size and laser beam energy was used to determine the ablation threshold [34,40,44,45]. The ablation threshold laser beam fluence of the ITO SL was 1.79 J/cm^2^. For Ag mid-layer thicknesses of 6, 13, and 16 nm, the calculated ablation threshold laser beam fluence were 0.39, 0.28, and 0.27 J/cm_2_, respectively. The significantly lower ablation thresholds of the ITO/Ag/ITO MLs were attributed to the higher thermal conductivity and absorption coefficient of the inserted Ag mid-layer [46]. Furthermore, the ablation threshold of the ITO/Ag/ITO MLs reduced slightly as Ag thickness increased.

Figure 4 shows the scanning electron microscopy (SEM) images of laser-ablated ITO/Ag/ITO MLs with different Ag thicknesses at a constant scanning speed of 500 mm/s and laser pulse energy of 67 μJ. As shown in Figure 4a, the ablated line width was approximately 58 μm for the ITO SL. For the ITO/Ag/ITO MLs with Ag thicknesses of 6, 13, and 16 nm, similar ablated line widths of 73–74 μm were measured, as shown in Figure 4b–d, respectively. Although the ITO/Ag/ITO MLs showed a wider ablation line width than the ITO SL without the inserted Ag mid-layer, the Ag thickness did not affect the laser-ablated line width, despite the different optical transmittances and absorption coefficients.

Figure 5 shows the SEM images of the spot-shaped laser-ablated patterns on ITO/Ag/ITO MLs with different Ag thicknesses at a fixed scanning speed of 2000 mm/s and a laser pulse energy of 97 μJ. The spot patterns were generated by the overlapping rate, i.e., the scanning speed of the laser beam divided by the repetition rate of the laser beam [20]. At a lower repetition rate, a higher laser pulse beam energy was obtained at the same scanning speed. Therefore, in cases involving a higher scanning speed and a lower repetition energy, spot-shaped ablated patterns were obtained instead of line-shaped ablated patterns (Figure 5). Figure 5a shows the ablated spot patterns on the ITO SL, whereas those on the ITO/Ag/ITO MLs for Ag thicknesses of 6, 13, and 16 nm are depicted in Figure 5b–d, respectively. The diameter of the ablated spot pattern on the ITO SL was 58 μm, and those on the ITO/Ag/ITO MLs with Ag thicknesses of 6, 13, and 16 nm were 83, 82, and 78 μm, respectively. This is similar to the results shown in Figure 4, where the ITO/Ag/ITO MLs showed a larger spot than the ITO SL, and the ablated spot sizes of the ITO/Ag/ITO MLs irrespective of the Ag thickness did not differ significantly. The ablated spot sizes of the ITO/Ag/ITO ML reduced slightly as the Ag thickness increased.

Figure 6 shows line widths of the laser-ablated ITO/Ag/ITO MLs as a function of the Ag mid-layer thickness at different laser pulse energies and different scanning speeds. The line widths of the ablated patterns increased with the laser pulse energy, as shown in Figure 6a. Compared with the ITO SLs, the ITO/Ag/ITO MLs showed a much wider ablated line width. However, the line widths did not change significantly with Ag thickness, as shown in Figure 4 and Figure 5. Additionally, Figure 6a shows the less dependence of the laser pulse energy on the ablated line widths as a result of the high thermal conductivity of the inserted Ag mid-layer. As the laser pulse energy decreased from 163 to 51 μJ, the laser ablated line width of the ITO SL decreased significantly from 58 to 21 μm by 63.8%. By contrast, the ablated line widths of the ITO/Ag/ITO MLs reduced by only approximately 15% to 20% irrespective of the Ag thickness. Figure 6b shows that the scanning speed affected only the ablated line width on the ITO SL and did not affect the ITO/Ag/ITO MLs. As the scanning speed increased from 50 to 2000 mm/s, the laser-ablated line width on the ITO SL reduced from 71 to 47 μm, i.e., by 33.8%. For the ITO/Ag/ITO MLs, the line width reduced by less than 5% irrespective of the Ag thickness. It is inferred that the scanning speed is closely associated with the overlapping rate of the laser beams, and that the overlap of the laser beam does not affect the laser ablated line width on the ITO/Ag/ITO MLs. This is likely because the inserted Ag mid-layer of a few nanometers thick has a much lower melting point than bulk Ag [47,48], causing it to melt and be removed easily during laser direct ablation.

Figure 7 shows the surface profiles of the laser-ablated ITO/Ag/ITO patterns with different Ag thicknesses at a scanning speed of 500 mm/s for a pulse energy of 97 μJ, as shown in Figure 6b. Considering the total thickness of the ITO/Ag/ITO MLs, it was confirmed that the ITO/Ag/ITO MLs were completely ablated by direct laser ablation. The edge areas of the ablated regions were investigated to confirm the complete laser direct ablation of the ITO/Ag/ITO MLs. Figure 8a,b shows the SEM images of the laser-ablated ITO SL and ITO/Ag/ITO ML, respectively, performed at a scanning speed of 1000 mm/s and laser pulse energy of 63 μJ. The relatively low overlapping rate of the laser beam at a high scanning speed enabled the groovy line on both the ITO SL and ITO/Ag/ITO MLs to be investigated. An observation of the edge part in the ablated region (shown in the tilted images) revealed a few ITO-like spikes that were generated by melting and vaporization during laser direct ablation, as shown in Figure 8a [49]. In the magnified view of Figure 8b, the larger ITO-like layers were rolled up by the thermal processes during laser direct ablation. Additionally, thin film residues were observed on the edge of the ablated region. Table 1 shows the material composition analyses of region 1 (ITO/Ag/ITO ML) and region 2 (the edge of the ablated region) via EDX. The results show that Sn and Ag were barely detected in region 2. It can be inferred that the upper and bottom ITO layers rolled up partially and remained on the edge of the ablated region because the thermal conductivity of ITO is considerably lower than that of Ag (see Appendix A). During laser direct ablation, the ITO layer is expected to be removed after the inserted Ag mid-layer is melted rapidly and vaporized.

For applications of the laser direct ablation of ITO/Ag/ITO MLs to optoelectronic and energy devices, the electrode patterns were fabricated under the same process conditions shown in Figure 7. Considering the realization of TCO electrodes, the ITO SL and ITO/Ag(6 nm)/ITO ML were selected as test electrodes based on the optical transmittance results depicted in Figure 2b. As shown in Table 2, the ITO/Ag(6 nm)/ITO ML electrode indicated a much higher electrical current density and lower resistance per unit length than the ITO SL electrode, despite the smaller cross section of the electrode caused by laser direct ablation.

### 3.2. Simulation Perspective

The simulation of transient temperature changes for ITO/Ag(16 nm)/ITO ML and ITO SL can support to elucidate the major contribution of the functional Ag layer in the laser ablation. Figure 9 and Figure 10 show the simulation results of the transient temperature distribution in the XZ-plane (depth direction) and XY-plane (z = 0) obtained from the irradiation of a single laser pulse (wavelength, 1064 nm; duration, 10 ns; laser beam fluence, 0.39 J/cm^2^) on the ITO/Ag(16 nm)/ITO ML and ITO SL. Glass was selected as the substrate under identical laser irradiation. First, when a Gaussian laser beam was incident in the depth direction, the maximum heating temperature difference between the ITO/Ag(16 nm)/ITO ML and ITO SL was approximately 800 K at 10 ns (Figure 9a,b). Furthermore, as the laser irradiation provided accumulated energy during 10 ns of irradiation, the transient temperature distribution in the depth direction (*Z*-axis) confirmed that a significant change in temperature occurred in the two cases. The increase in temperature in the Ag mid-layer case (Figure 9c) was greater than that of the ITO SL (Figure 9d) at all transient times. When the Ag layer was inserted in ITO layers, it could help to absorb more laser rather than the transmitted portion and provide more converted thermal energy to the adjacent ITO layers through the out-of-plane heat transfer from the Ag to ITO layer. Furthermore, while the high thermal conductivity of the Ag layer can promote the in-plane heat transfer inside the Ag layer, the elevated temperature of the Ag layer may result in transferring the thermal energy to the neighboring ITO layer through the out-of-plane heat transfer. Consequently, the high laser absorption coefficient (α = 1.03 × 10^5^ cm^−1^ at 1064 nm) and high thermal conductivity (429 W/m·K) of the Ag layer enabled more intensively focused energy on the target layer, thereby resulting in a significant temperature gradient and enhanced fabrication efficiency.

The temperature distribution in the XY plane shows more intensively focused photothermal energy in the ITO/Ag/ITO ML (Figure 10a) compared with in the ITO SL (Figure 10b). The overall temperature range of the ITO/Ag/ITO ML (Figure 10c) as a result of laser beam irradiation (80 µm in width) was considerably higher than that of the ITO SL (Figure 10d). Since the sharp gradient of temperature indicates the threshold regime for the laser ablation, it was comparable to the ablated spot in the experimental measurements (Figure 6 and Figure 7). Moreover, because the coverage ranges of the laser beam width were similar for the two cases, the higher temperature range might indicate a significant temperature gradient in the same area, and more precise patterning would be obtained for the processing conditions. Therefore, the insertion of a highly conductive Ag layer affords enhanced patterning in terms of width through laser direct ablation. Moreover, the transient temperature changes within 0 to 10 ns based on the thickness change of the Ag layer in the layered structures are shown in Figure 10e. The common feature for all Ag thicknesses was the increasing temperature trend during irradiation. Meanwhile, as the Ag thickness increased from 6 to 16 nm, the absorbed photothermal energy increased, owing to the larger thermal mass for an effective photothermal energy conversion. However, using a larger thermal mass by increasing the thickness of the highly conductive Ag layer can result in more dissipation of converted thermal energy in the XY plane, and the widely distributed thermal energy may result in an increased line width of the laser-ablated pattern, as shown in Figure 6. This indicates that the optimal Ag thickness, which can absorb the appropriate photothermal energy but prohibit wide thermal energy dissipation, is crucial for generating a precise line width.

## 4. Conclusions

Although issues of general concern about possible disadvantages, such as more complicated fabrication process with increasing cost, still remain, in this work, we successfully demonstrated the fabrication of ITO/Ag/ITO MLs for transparent conductive oxide electrode applications with a higher *FoM* compared with that of ITO SL. To analyze the effects of the Ag mid-layer on the laser direct ablation-assisted patterning of OMO ML films, a Nd:YVO_4_ laser was used in the direct etching of ITO/Ag/ITO MLs deposited for various Ag thicknesses using an in-line sputtering method. In a comparison study with ITO SL processing, the results of the laser direct ablation of ITO/Ag/ITO MLs were investigated and evaluated by varying the laser direct ablation processing conditions, such as the scanning speed and laser pulse energy of the laser beam. By inserting the Ag mid-layer between the top and bottom ITO layers, the widths of the laser-ablated ITO/Ag/ITO ML patterns became much wider than those of the ITO SL pattern. However, compared with the ITO SL, the laser-ablated line width of the ITO/Ag/ITO MLs was not affected by the laser beam conditions, such as laser pulse energy or overlapping rate. In addition, the thickness of the inserted Ag mid-layer was not significantly affected by the pattern width. This is attributable to the easy melting and evaporation of the Ag mid-layer during ablation, which was enabled by the considerable melting-point depression as the layer thickness decreased. The computational simulation supported the experimental results obtained in this study, thereby demonstrating the pivotal role of the Ag mid-layer in the effective absorption and focusing of photothermal energy. Using the optimized laser direct ablation process conditions, ITO/Ag/ITO ML electrodes with a higher conductivity than the ITO SL electrode (e.g., each current density obtained at 1 V was 0.29 × 10^4^ and 2.98 × 10^4^ A/cm^2^ for ITO SL and ITO/Ag(6 nm)/ITO ML electrodes, respectively) can be realized without significant degradation in the optical transmittance in the visible light regime, where each value of *FoM* obtained was 1.88 × 10^−3^ and 2.74 × 10^−3^ for ITO SL and ITO/Ag(6 nm)/ITO ML electrodes, respectively. The results obtained are attributable to the high thermal conductivity (e.g., each value for ITO and Ag is 11.5 and 429 W·m^−1^·K^−1^ respectively) and absorption coefficient of the inserted Ag mid-layer (e.g., each value for ITO and Ag is 4 × 10^5^ and 1.03 × 10^7^ m^−1^, respectively), which are also shown in the Appendix A. This study provides critical insights into the development of the design principle of high-performance TCO electrodes for improving the electrical and optical performances of OMO electrodes for various optoelectronic and energy devices. In future studies, the material parameters of thin metal mid-layers, such as the control of interfacial roughness, composition and degree of defects, and submicron-level alloying, should be modified to further enhance the overall performance of device applications.

## Figures and Tables

**Figure 1 materials-14-05136-f001:**
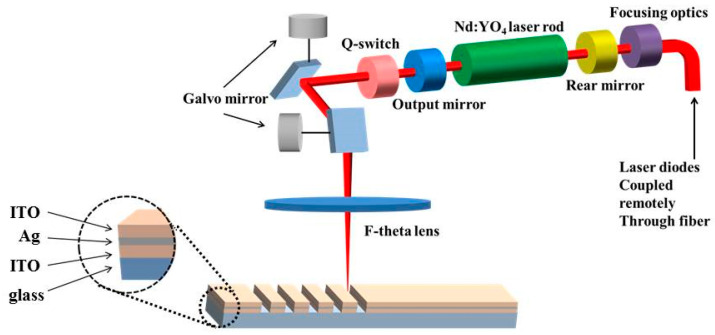
Schematic diagram of experimental setup for laser direct ablation of ITO/Ag/ITO MLs on glass substrates.

**Figure 2 materials-14-05136-f002:**
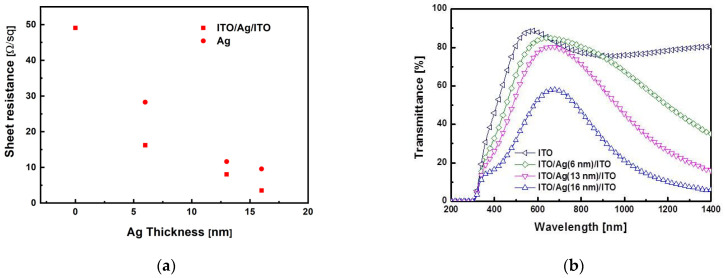
Electrical and optical properties of prepared ITO/Ag/ITO MLs with different Ag thickness: (**a**) sheet resistances and (**b**) transmittances.

**Figure 3 materials-14-05136-f003:**
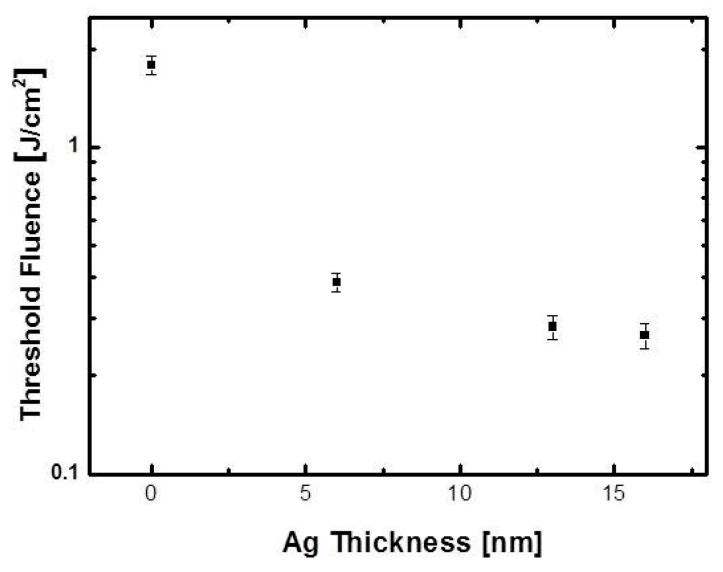
Laser direct ablation thresholds of ITO/Ag/ITO MLs with different Ag thicknesses.

**Figure 4 materials-14-05136-f004:**
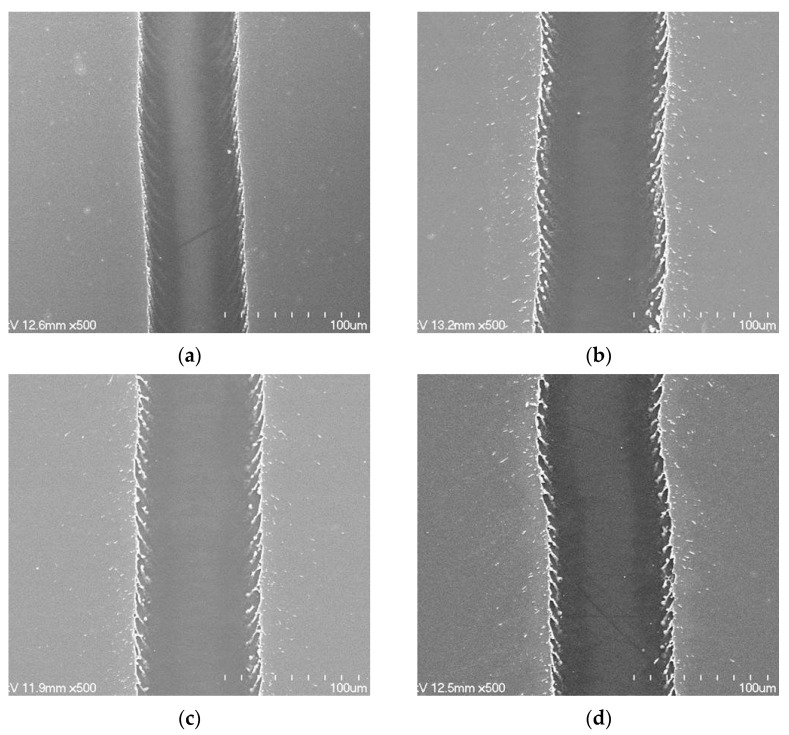
SEM images of laser-ablated ITO/Ag/ITO MLs with different Ag thicknesses of (**a**) 0, (**b**) 6, (**c**) 13, and (**d**) 16 nm at a scanning speed of 500 mm/s and laser pulse energy of 67 μJ.

**Figure 5 materials-14-05136-f005:**
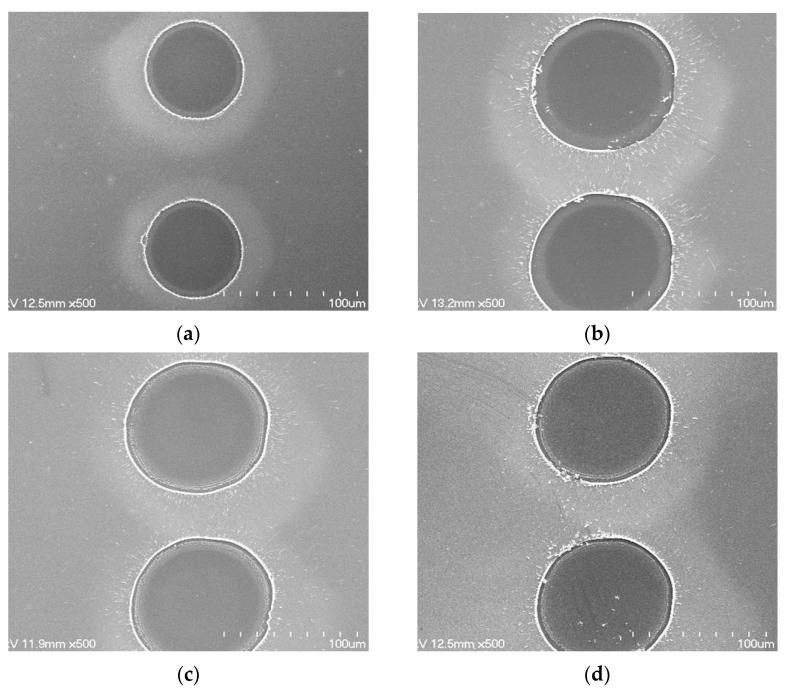
SEM images of laser-ablated ITO/Ag/ITO MLs with different Ag thicknesses of (**a**) 0, (**b**) 6, (**c**) 13, and (**d**) 16 nm at scanning speed of 2000 mm/s and laser pulse energy of 97 μJ.

**Figure 6 materials-14-05136-f006:**
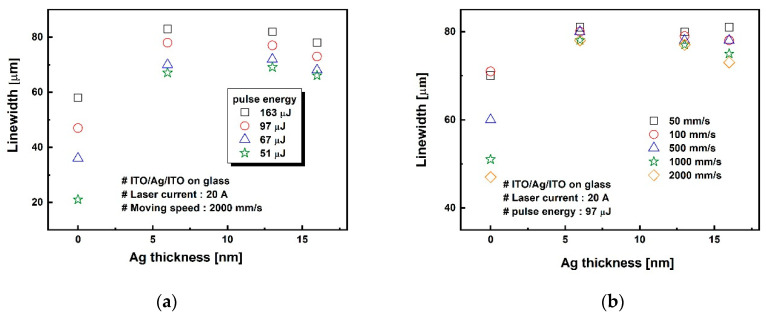
Lines widths of laser-ablated ITO/Ag/ITO MLs as a function of Ag thickness with (**a**) different laser pulse energies (scanning speed, 500 mm/s) and (**b**) different scanning speed (laser pulse energy, 97 μJ).

**Figure 7 materials-14-05136-f007:**
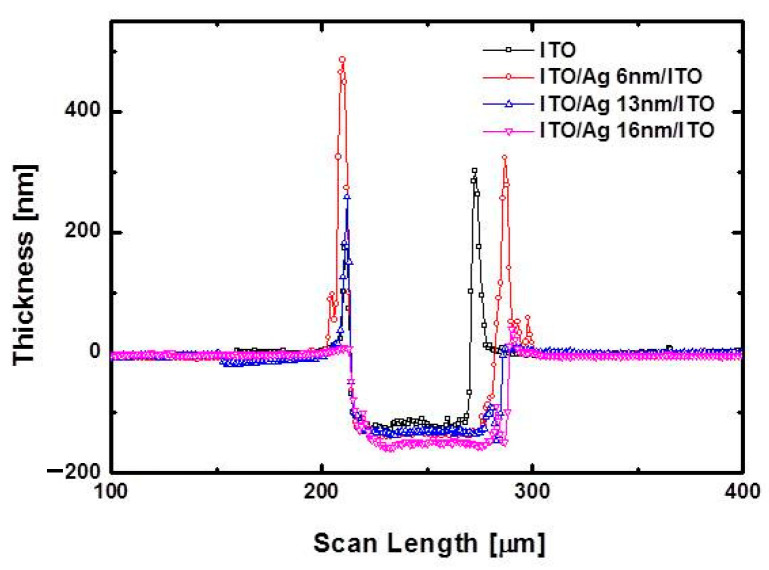
Surface profiles of laser-ablated ITO/Ag/ITO MLs with different Ag thicknesses at scanning speed of 500 mm/s for pulse energy of 97 μJ.

**Figure 8 materials-14-05136-f008:**
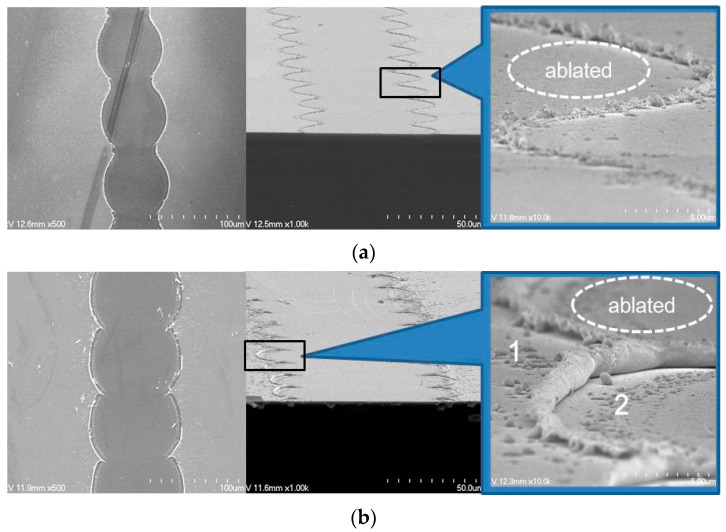
SEM images of laser-ablated ITO/Ag/ITO MLs with different Ag thickness of (**a**) 0, and (**b**) 13 nm at scanning speed of 1000 mm/s and laser pulse energy of 163 μJ.

**Figure 9 materials-14-05136-f009:**
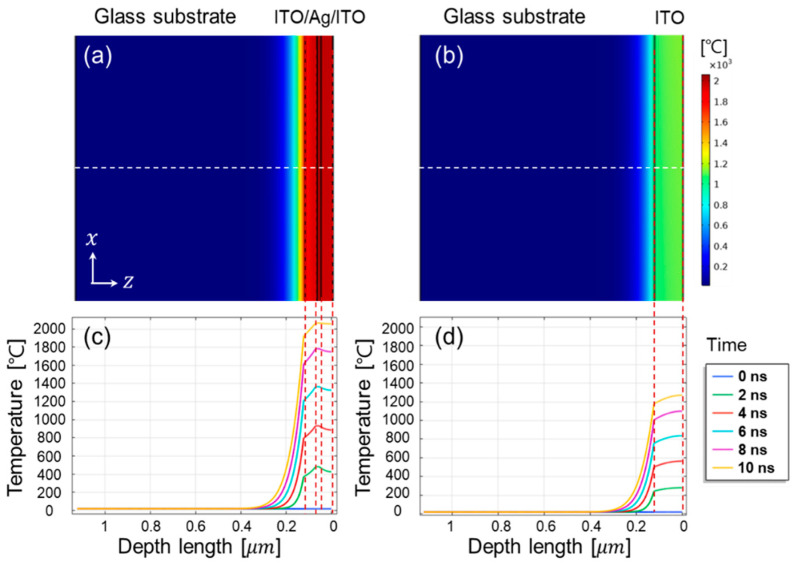
Transient temperature analysis in XZ-plane of ITO (53 nm)/Ag (16 nm)/ITO (53 nm) on 1 µm thick glass substrate (**a**,**c**) and of ITO SL on the same glass substrate (**b**,**d**). Laser irradiation (λ, 1064 nm) in single pulse (duration of 10 ns) of energy fluence 0.39 J/cm^2^.

**Figure 10 materials-14-05136-f010:**
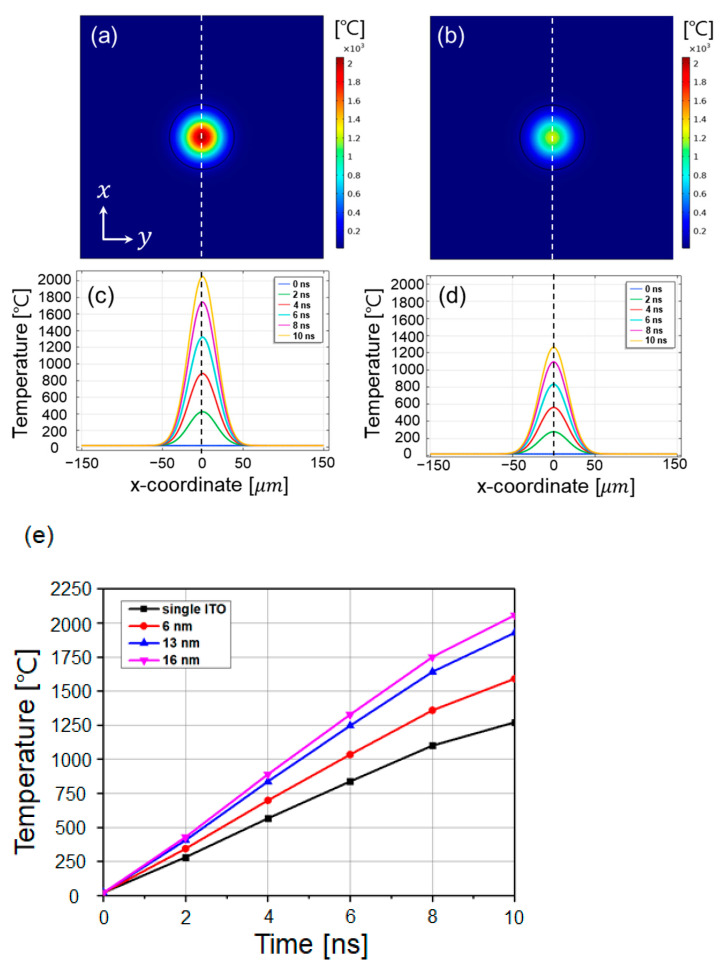
Transient temperature analysis in XY-plane (z = 0). Temperature distributions of ITO (53 nm)/Ag (16 nm)/ITO (53 nm) on 1 µm thick glass substrate (**a**,**c**) and of ITO SL on the same glass substrate (**b**,**d**). Transient temperature variations for every 2 ns presented by different color lines. (**e**) Comparison of temperature variations on central point, (x, y, z) = (0, 0, 0) during entire pulse duration (0 to 10 ns) based on variation of silver layer thickness (6, 13, and 16 nm).

**Table 1 materials-14-05136-t001:** EDX results of laser-ablated ITO/Ag/ITO ML patterns shown in Figure 8.

					[Units: at%]
	In	Sn	Ag	O	Si
(1) Remaining region	33.00	4.06	2.29	35.32	18.70
(2) Ablated region	3.29	0	0.79	50.24	31.66

**Table 2 materials-14-05136-t002:** Electrical characteristics of laser-ablated ITO SL and ITO/Ag/ITO ML electrodes.

	ITO/Ag(0 nm)/ITO	ITO/Ag(6 nm)/ITO
Width of electrode (b) [10^−4^·cm]	149.09	123.96
Thickness of electrode (t) [10^−4^·cm]	0.092	0.098
Current density at 1 V [10^4^·A/cm^2^]	0.29	2.98
Resistance per unit length [kΩ/cm]	6.475	0.693

## Data Availability

The data presented in this study are available on request from the corresponding author. The data are not publicly available due to the privacy of this research.

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
