# Peer review of "Effects of Insertion of Ag Mid-Layers on Laser Direct Ablation of Transparent Conductive ITO/Ag/ITO Multilayers: Role of Effective Absorption and Focusing of Photothermal Energy"

_materials, 2021, doi:10.3390/ma14185136_

Round 1
Reviewer 1 Report
Results for optical and electrical properties have already been investigated in 2009 (e.g. https://doi.org/10.1016/j.optcom.2008.10.075), laser processing (ablation) has no influence on these properties. Thus, it is not clear where the novelty of this work might be.
Studies on laser ablation are new to the point, that they haven´t been done with a ns-laser with the respective wavelength. The reason for this is the ablation-quality and the resulting burr on the edges, which is also very present in this work (burr exhibits larger dimensions than the layer thickness). As a consequence, the applicability in optoelectronic devices is very limited. There are only very limited new insights (as mentioned by the authors) on the ablation process, just additional proofs of already established theories.
Direct laser ablation of IMI-layers in general has been done before:
- KIM, Hyo-Joong, et al. Direct laser patterning of transparent ITO–Ag–ITO multilayer anodes for organic solar cells. Applied surface science, 2015
- KUBIŠ, Peter, et al. Patterning of OPV modules by ultra-fast laser. In: Laser Processing and Fabrication for Solar, Displays, and Optoelectronic Devices III. International Society for Optics and Photonics, 2014
In terms of cientific methods there is at least one question regarding Figure 6. It is unclear if the lines represent a functional fit? (If so, it´s not a very good fit). In fact, they are not helpful for a better visual clarification, so they might be better removed.
The determination of the ablation threshold needs to be clarified. Liu-plot for the determination would make it more comprehensible or at least the value for ω0. Additionally, 4 data points (there are not more pulse energies mentioned) for this kind of fit are not appropriate.
Reviewer 2 Report
An ITO surface was used as template for the deposition of silver using a sputtering technique and then the surface was ablated under pulsed laser conditions. The new composites material was organized at three different values of thickness in order to evaluate the optoelectronic of composite. Follow several characterisations by SEM techniques.
Line 86. Is suggested to Introduce the condition of ultrasonically, frequency, power and instrument used.
Line 130. 2.4. Computational simulation. It is indicated to extend this part without remand the reader to the supplement information.
Reviewer 3 Report
The paper entitled "Effects of insertion of Ag mid-layers on laser direct ablation of transparent conductive ITO/Ag/ITO multilayers: role of effective absorption and focusing of photothermal energy” by Choi et al deals with the study of ablation lines in ITO/Ag/ITO layers using a Nd:YVO4 laser. The influence of the intermediate layer on the ablation properties is studied.
Results provided in this review paper are in the scope of the Materials journal. These are interesting, and can find scientific and industrial applications; however, after reading the paper, I have some comments about it:
In general, this paper is interesting and it is well-written. The only main issue in this work is related with the thermal simulation. Is required a thermal simulation in this case? Simulations are normally done to explain results, however, in this case, the performed simulations do not provide big explanations on what is happening, and the results provided by these are evident. Then, I would recommend the removal of this section or try to use it better to explain the results.
COMMENTS:
- (Page 1) ML acronym has not defined. Please, define it the first time is used in the main text.
- (Page 2, Line 89, Experimental) Please add the error to the measurement of the thickness for ITO (53 nm).
- (Page 3, Experimental) Please, add the beam quality, TEM mode (TEM00 in this case), focal length and beam spot diameter to the laser experimental processing conditions.
- (Page 4, Results) Please, add error bars to the results of the sheet resistance given in Fig. 2a. Why was an interpolation line added to these results? No mention to the interpolation equation is given in the main text.
- (Page 5, Results) Authors attribute the lower ablation threshold to the higher thermal conductivity and absorption coefficient of the inserted Ag mid-layer. Could be the reduction of the transmissivity of the laser radiation (as seen in Fig. 2b) also an explanation of this reduction in the ablation threshold?
- (Pages 5-6, Results) It seems that the ablated lines or spots are cleaner when no Ag layer is present. Why? Due to a higher absorption and a more energetic ablation of the material?
- (Page 6, Results) Please, add the errors for the ablated spot diameters.
- (Page 7, Results) Please, add error bars to the data shown in Fig. 6. Again, why an interpolation line was added if the equation of the line is not given? Authors should leave the data as isolated points or connected by straight lines. Interpolations are usually given to provide estimations that should be corroborated by confirmation experiments.
- (Page 8, Results) Which are the units for the EDX results? at%, wt%? Please, clarify it in the table 1.
- (Page 9, Lines 410-414, Results) The higher absorption produces a larger absorption of laser radiation and conversion into heat. The higher thermal conductivity produces a larger diffusion of produced heat during the thermalization of the laser radiation. None of these two parameters “enables a more intensively focused energy on the target layer”. Focusability is not related to the parameters of the substrate, but with the focusing system and properties/characteristics of the laser radiation.
- (Pages 9-11) Authors should ask about the necessity of simulation and on the validity of the results. Is needed the thermal simulation in this case? How was validated the simulation? Authors could compare the results of the thermal field with the ablated spots.
Reviewer 4 Report
The submitted manuscript studies the properties of thin metal mid-layers with addition of silver mid-layers and I can comment upon it as follows:
- Addition of an Ag mid-layer improves certain properties of ITO/Ag/ITO multilayers (it should be pointed out, which particular ones) and compromise some others (maybe, transparency and so forth). This should be discussed from the viewpoint of the net effect.
- It is necessary to substantiate the choice of Ag for introduction between the top and bottom ITO layers and discuss other (for example, less expensive) materials for the same purpose as that of Ag.
- The Conclusion uses such qualitative descriptions as “much higher conductivity”, “high thermal conductivity”. These parameters should be illustrated by their quantitative values and compared to those of films without Ag.
The submitted manuscript may be published in Materials after the Authors address the provided concerns in a further version.
Round 2
Reviewer 1 Report
The authors have provided sufficent response and a revised version which describes the originality of the manuscript and the comprehensive approach to reveal the functional role of the mid Ag layer in a much better way. This has been the major issue in the first review. The other two aspects mentioned in the first review have been minor aspects and have been responded appropriately, as well.
Reviewer 4 Report
The earlier raised concerns were partially addressed, but the manuscript is still difficult to read, especially for researchers who are not deeply involved in this topic. It would be highly desirable to have explicit answers to questions that suggest themselves strongly when reading the text:
- Once again, the Authors need to list not exclusively the advantages of the proposed design of ITO/Ag/ITO multilayers, but also its disadvantages (e.g. more complicated fabrication process, cost, &c). Such an unbiased evaluation of the proposed design should be given both in the Abstract and Conclusion.
- The Conclusion still remains to a large degree qualitative and provides almost no data (only references have been added to Table 2 and Supplementary Materials). It is necessary to add explicitly quantitative parameters that could illustrate the advantages (and drawbacks) of the proposed design.
If these issues are resolved in a further revision of the manuscript, it may be published in Materials.
Round 3
Reviewer 4 Report
After revision, the manuscript was improved, and the advantages of the proposed composite material are more understandable. I have no more essential concerns, the manuscript may be now published in its latest version.